# Attenuation parameter and liver stiffness measurement using FibroTouch vs Fibroscan in patients with chronic liver disease

Ying Zhuang Ng[1], Lee Lee Lai[1,2], Sui Weng Wong[1], Siti Yatimah Mohamad[1], Kee Huat Chuah[1], Wah Kheong Chan[1] *

1 Faculty of Medicine, Department of Medicine, Gastroenterology and Hepatology Unit, University of Malaya, Kuala Lumpur, Malaysia, 2 Department of Nursing Science, University of Malaya, Kuala Lumpur, Malaysia

* wahkheong2003@hotmail.com

## Abstract

### Background & aim

We studied FibroTouch (FT) and Fibroscan (FS) examination results and their repeatability when performed by healthcare personnel of different background.

### Methods

FT and FS examinations were performed on patients with chronic liver disease by two operators, a doctor and a nurse, twice on each patient, at two different time points, independent of each other.

### Results

The data for 163 patients with 1304 examinations was analyzed. There was strong correlation between FT and FS for attenuation parameter (Spearman's rho 0.76, p<0.001) and liver stiffness measurement (LSM) (Spearman's rho 0.70, p<0.001). However, FT produced higher value at lower attenuation parameter and LSM, and lower value at higher attenuation parameter and LSM. There was substantial agreement when using 15kPa LSM cut-off, but only moderate agreement when using 10kPa and 20kPa LSM cut-offs and 248dB/m, 268dB/m and 280dB/m attenuation parameter cut-offs. The IQR for attenuation parameter and IQR/median for LSM were significantly lower for FT compared with FS (4dB/m vs 27dB/m, p<0.001, and 10 vs 12, p<0.001, respectively). The intra- and inter-observer reliability of attenuation parameter and LSM using FT and FS were good to excellent with intraclass correlation coefficients 0.89–0.99. FT had shorter examination time (33s vs 47s, p<0.001) and less invalid measurements (0 vs 2, p<0.001).

### Conclusion

Measurements obtained with FT and FS strongly correlated, but significant differences in their absolute values, consistency, examination time and number of invalid measurements

**Data Availability Statement:** All relevant data are within the paper and its Supporting Information files.

**Funding:** The authors received no specific funding for this work.

**Competing interests:** The authors have declared that no competing interests exist.

were observed. Either device can be used by healthcare personnel of different backgrounds when sufficiently trained.

## Introduction

Chronic liver disease causes major health burden with increasing morbidity and mortality worldwide [1]. Early identification of liver fibrosis regardless of aetiology is critical in the management of chronic liver disease as it is one of the most important predictors of long-term outcome. The gold standard for staging liver fibrosis remains histopathological examination of a liver biopsy specimen. However, due to the invasive nature of the liver biopsy procedure, many non-invasive diagnostic techniques have been developed. Transient elastography is a useful non-invasive tool for the assessment of liver fibrosis [2]. Fibroscan (FS) is a vibration-controlled transient elastography device that estimates liver stiffness for the diagnosis of liver fibrosis; additionally, controlled attenuation parameter (CAP), which is derived from the same radiofrequency data for liver stiffness measurement, is used for the diagnosis of hepatic steatosis [3]. It has been validated in numerous studies, is widely accepted, and has been incorporated into major guidelines [4–6]. In 2013, FibroTouch (FT), also a vibration-controlled transient elastography device, was introduced. Similar to FS, liver stiffness measurement (LSM) and the attenuation parameter (called ultrasound attenuation parameter, UAP) obtained using FT can be used for the diagnosis of liver fibrosis and hepatic steatosis [7]. The primary aim of this study was to compare LSM and attenuation parameter obtained using FT and FS in patients with chronic liver disease of various aetiologies. Our secondary aims were to determine the intra-observer and inter-observer variability of the two modalities in healthcare personnel of different backgrounds, and to compare the time taken to complete an examination with FT and FS.

## Methods

### Study design

This is a cross-sectional study of consecutive adult patients who were scheduled for FS examination at the University of Malaya Medical Centre, Kuala Lumpur, Malaysia between September 2019 to March 2020. The study conformed to the ethical guidelines of the 1975 Declaration of Helsinki and ethical approval was obtained from the University Malaya Medical Centre Medical Research Ethics Committee prior to commencement (MECID: 201982–7708). Adults with chronic liver disease of various aetiologies were included in this study. The exclusion criteria were patients younger than 16 years old, malignancy, ascites, extrahepatic cholestasis and pregnancy. All participating subjects provided written informed consent.

Demographic, anthropometric, clinical, and laboratory data were recorded using a standard protocol. BMI was calculated by dividing weight in kilogram by the square of height in meter. Subjects with BMI $\geq 25$ kg per $m^2$ were considered as obese [8]. Waist circumference was measured at the midpoint between the lowest margin of the least palpable rib and the top of the iliac crest in the standing position. Central obesity was defined as a waist circumference $\geq 90$ cm in men and $\geq 80$ cm in women [9]. Venous blood was drawn after overnight fasting for fasting blood glucose, liver and lipid profile within 6 months before or after elastography examination.

## Transient elastography

The transient elastography devices used in this study were Fibroscan 502 Touch (Echosens, France) and FibroTouch FT100 (Wuxi Hisky Medical Technology Co. Ltd, Wuxi, China). Examinations using both devices were performed by two operators, twice on each patient, at two different time points, independent of each other. One operator was a doctor (operator 1, N. Y. Z.), while the other operator was a nurse (operator 2, L. L. L.).

## Fibroscan examinations

FS was performed after ≥2 hours of fasting. Each of the operators have performed ≥100 examinations prior to commencement of this study. An examination was considered successful if there were ≥10 valid measurements and reliable if the interquartile range (IQR)/median for liver stiffness measurement was ≤30% or if the liver stiffness measurement was <7.1 kPa when the IQR/median was >30% [10,11]. An examination was considered unsuccessful if <10 valid measurements were obtained after 30 attempts [12]. The use of Fibroscan probe, M or XL, was based on computer recommendation. The time taken to complete an examination, from the time the probe was first placed onto the patient until the examination was completed or considered unsuccessful, was recorded [12].

## FibroTouch examinations

FT was also performed after ≥2 hours of fasting. Both operators have not performed FT previously and have received training for the purpose of this study. Following training, each of the operators have performed ≥100 examinations prior to commencement of this study. Similar to FS, an examination was considered successful if there were ≥10 valid measurements and reliable if the IQR/median for liver stiffness measurement was ≤30% or if the liver stiffness measurement was <7.1 kPa when the IQR/median was >30%. An examination was also considered unsuccessful if <10 valid measurements were obtained after 30 attempts. The time taken to complete an examination, from the time the probe was first placed onto the patient until the examination was completed or considered unsuccessful, was recorded.

## Statistical analysis

Data were analysed using a standard statistical software program, SPSS 25.0 (SPSS Inc, Chicago, IL, USA). Continuous variables were expressed as mean ± standard deviation or median (IQR), where appropriate. Categorical variables were expressed as percentages. Spearman correlation coefficient rho and scatter plot were used to compare LSM and attenuation parameters obtained using FS and FT whereby | rho | ≤ 0.30 was considered weak correlation, 0.30 < | rho | < 0.70 moderate correlation, and | rho | ≥ 0.70 strong correlation. The median attenuation parameter, IQR of attenuation parameter, LSM, and IQR/median of LSM, time taken to complete an examination and number of invalid measurements using the two devices were compared using Wilcoxon test. Significance was assumed when p <0.05. Bland Altman plot was used to analyse the agreement between measurements obtained using the two devices. The agreement in diagnosis of liver fibrosis and hepatic steatosis using LSM and attenuation parameter obtained using the two devices were analysed with Cohen's Kappa (K) value whereby K ≤0 was considered no agreement, 0.01–0.20 slight agreement, 0.21–0.40 fair agreement, 0.41–0.60 moderate agreement, 0.61–0.80 substantial agreement, 0.81–0.99 almost perfect agreement. The LSM and attenuation parameter cut-offs for diagnosis were based on previously published studies for FS. The cut-offs for attenuation parameter used for the diagnosis of steatosis grades >S0, >S1 and >S2 were 248 dB/m, 268 dB/m and 280 dB/m,

respectively [13]. The LSM cut-offs used were <10 kPa, >15 kPa and >20 kPa, indicative of unlikely compensated advanced chronic liver disease, likely compensated advanced chronic liver disease and likely clinically significant portal hypertension, respectively [4,14]. In addition, subgroup analyses were performed: (1) using LSM cut-offs 8 kPa and 11 kPa for the diagnosis of significant fibrosis and cirrhosis, respectively, for patients with chronic hepatitis B [15], and (2) using LSM cut-offs 7 kPa an 10.3 kPa for the diagnosis of significant fibrosis and cirrhosis, respectively, for patients with NAFLD [16]. Further analyses was performed to look at the diagnostic accuracy of LSM obtained by FT, using LSM obtained by FS as the reference standard for the aforementioned diagnostic goals. Diagnostic accuracy was determined based on area under receiver operating characteristic curve (AUROC), which was interpreted as follows: 0.90–1.00 = excellent, 0.80–0.90 = good, 0.70–0.80 = fair, < 0.70 = poor. The optimal cut-off for LSM obtained by FT for a particular diagnostic goal was the LSM value that provided the highest sum of sensitivity and specificity for that diagnostic goal. The sensitivity, specificity, positive predictive value, negative predictive value, and accuracy of LSM obtained by FT for a particular diagnostic goal were determined based on the optimal cut-off for that diagnostic goal. Intra-observer and inter-observer reliability were analysed using intraclass correlation coefficient. Intraclass correlation coefficient (ICC) was interpreted as follows: 0.90–1.00 = excellent, 0.80–0.90 = good, 0.70–0.80 = fair, and 0.70 = poor.

## Results

A total of 1336 transient elastography (TE) examinations with FT and FS were performed on 167 patients. There were no adverse events from both procedures. One patient was excluded due to age less than 16 years old. Three patients were excluded from the analysis due to failed or unreliable TE results. There was failure to obtain 4 sets of reliable LSM in two patients using FS and in one patient using both FT and FS. The rate of successful and reliable examination using FT and FS was 99% and 98%, respectively. After exclusion of the aforementioned patients, 163 patients with 1304 examinations were included for analysis. Only 6 (1%) FS examinations were performed using the XL probe. The rest of the FS examinations were performed using the M probe. Patients characteristics are summarized in **Table 1**. The mean age of the study population was 57 ± 14 years and 52% were males. The mean BMI was 25.7 ± 4.5 kg per m$^2$ and ranged between 15 kg per m$^2$ and 40.4 kg per m$^2$. The waist circumference was 89.6 ± 12.7 cm and ranged between 59.0 cm and 122.5 cm. The prevalence of obesity and central obesity was 52% and 62%, respectively. The main aetiology of chronic liver disease was non-alcoholic fatty liver disease (NAFLD, 48.5%), followed by chronic hepatitis B (46%), chronic hepatitis C (0.6%) and other aetiologies (4.9%).

### Attenuation parameters

There was a strong correlation between FT and FS for attenuation parameter (Spearman's rho = 0.76, p <0.001) (**Fig 1A**). However, FT tended to produce higher value at lower attenuation parameters but lower value at higher attenuation parameters (**Fig 1B**). Attenuation parameter obtained using FT and FS were significantly different overall, FT being higher at lower attenuation parameter and lower at higher attenuation parameter (**Table 2**). The IQR for attenuation parameter obtained using FT was significantly lower compared with the IQR for attenuation parameter obtained using FS (**Table 2**). Bland–Altman analysis for attenuation parameters revealed a mean difference of -4.45 dB/m between attenuation parameters obtained with FT and FS, with the lower 95% limit of agreement being −86.28 dB/m and the upper 95% limit of agreement being 77.38 dB/m (**Fig 2**). There was only moderate agreement

**Table 1. Baseline characteristics of study population.**

| Characteristics | Overall population (n = 163) |
|---|---|
| Age, years | 57 ± 14 |
| Male, n (%) | 85 (52) |
| Body mass index, kg/m$^2$ | 25.7 ± 4.5 |
| Waist circumference, cm | 89.6 ± 12.7 |
| Obesity, n (%) | 85 (52) |
| Central obesity, n (%) | 101 (62) |
| Diabetes mellitus, n (%) | 50 (31) |
| Hypertension, n (%) | 52 (32) |
| Dyslipidaemia, n (%) | 64 (39) |
| Albumin, g/L | 39 (36–41) |
| Bilirubin, umol/L | 12 (9–16) |
| ALT, U/L | 34 (21–52) |
| ALP, U/L | 76 (63–93) |
| GGT, U/L | 33 (20–68) |
| Total Cholesterol, mmol/L | 4.6 (3.9–5.4) |
| HDL Cholesterol, mmol/L | 1.3 (1.1–1.5) |
| LDL Cholesterol, mmol/L | 2.6 (2.1–3.3) |
| Triglyceride, mmol/L | 1.4 (0.9–1.9) |
| Fasting blood glucose, mmol/L | 5.4 (4.9–6.8) |
| Indications of LSM: | |
| Chronic hepatitis B, n (%) | 75 (46) |
| Chronic hepatitis C, n (%) | 1 (0.6) |
| Non-alcoholic fatty liver, n (%) | 79 (48.5) |
| Others, n (%) | 8 (4.9) |

Continuous variables are presented as mean ± standard deviation or median (interquartile range).

Obesity was defined as body mass index, ≥25 kg per m$^2$.

Central obesity was defined as waist circumference >90cm for men and >80cm for women.

**ALT**, alanine aminotransferase; **ALP**, alkaline phosphatase; **GGT**, γ-glutamyl transpeptidase; **LDL**, low density lipoprotein; **HDL**, high density lipoprotein; **LSM**, liver stiffness measurement.

between FT and FS with Cohen's kappa value of 0.57, 0.56 and 0.58 when using attenuation parameter 248 dB/m, 268 dB/m and 280 dB/m as cut-off, respectively.

There was a strong correlation between operator 1 and 2 for attenuation parameter measured using FT (Spearman's rho = 0.85, p <0.001) and FS (Spearman's rho = 0.83, p <0.001) (**S1A and S1B Fig**). The intra-observer reliability of UAP obtained using FT by operator 1 and 2 were excellent with ICC 0.95 (95% CI, 0.93–0.96) and ICC 0.94 (95% CI, 0.92–0.96), respectively. The intra-observer reliability of CAP obtained using FS by operator 1 and 2 were also excellent with ICC 0.94 (95% CI, 0.92–0.96) and ICC 0.90 (95% CI, 0.86–0.93), respectively. The inter-observer reliability of UAP using FT was excellent with ICC 0.92 (95% CI, 0.90–0.93), whereas the inter-observer reliability of CAP obtained using FS was good with ICC 0.89 (95% CI, 0.87–0.91).

## Liver stiffness measurement

There was also a strong correlation between FT and FS for LSM (Spearman's rho = 0.70, p <0.001) (**Fig 3A**). However, FT tended to produce lower value at higher LSM (**Fig 3B**). LSM

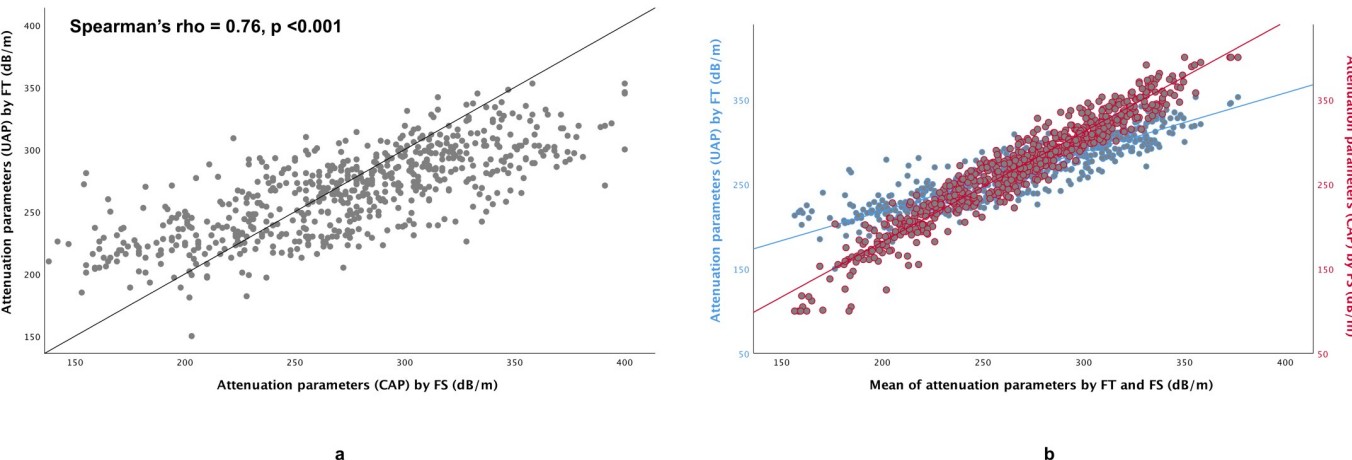

**Fig 1.** Scatter plots for (**a**) attenuation parameters obtained using FibroTouch (FT) vs Fibroscan (FS), and (**b**) attenuation parameters obtained using FT and FS vs mean of attenuation parameters obtained using FT and FS.

obtained using FT and FS were significantly different overall, being higher at lower LSM and lower at higher LSM (**Table 2**). The IQR/median for LSM obtained using FT was significantly lower compared with the IQR/median for LSM obtained using FS (**Table 2**). Bland–Altman analysis for LSM revealed a mean difference of 0.91 kPa between liver stiffness measurement

**Table 2. Comparison of attenuation parameter and liver stiffness measurement obtained using FibroTouch and Fibroscan.**

| | FibroTouch | Fibroscan | p |
|---|---|---|---|
| **Overall, n = 652** | | | |
| Attenuation parameter, dB/m | 268 (237–292) | 274 (230–313) | <0.001 |
| IQR, dB/m | 4 (2–7) | 27 (19–37) | <0.001 |
| **Mean attenuation parameter <250 dB/m, n = 218** | | | |
| Attenuation parameter, dB/m | 227 (216–242) | 210 (179–231) | <0.001 |
| IQR, dB/m | 5 (3–8) | 32 (22–47) | <0.001 |
| **Mean attenuation parameter 250–300 dB/m, n = 267** | | | |
| Attenuation parameter, dB/m | 272 (259–284) | 279 (266–297) | <0.001 |
| IQR, dB/m | 5 (2–7) | 27 (20–37) | <0.001 |
| **Mean attenuation parameter >300 dB/m, n = 167** | | | |
| Attenuation parameter, dB/m | 303 (293–317) | 337 (320–353) | <0.001 |
| IQR, dB/m | 1 (1–3) | 22 (16–30) | <0.001 |
| **Overall, n = 652** | | | |
| LSM, kPa | 8.3 (6.4–11.5) | 6.8 (5.1–9.3) | <0.001 |
| IQR/median, % | 10 (6–16) | 12 (9–18) | <0.001 |
| **Mean LSM <15 kPa, n = 598** | | | |
| LSM, kPa | 7.8 (6.3–10.6) | 6.5 (5.0–8.6) | <0.001 |
| IQR/median, % | 10 (6–17) | 12 (9–18) | <0.001 |
| **Mean LSM ≥15 kPa, n = 54** | | | |
| LSM, kPa | 18.1 (15.3–19.8) | 22.8 (17.9–27.8) | <0.001 |
| IQR/median, % | 7 (5–13) | 13 (9–18) | 0.001 |

n represents the number of examinations using each of the two devices.

**LSM**, liver stiffness measurement.

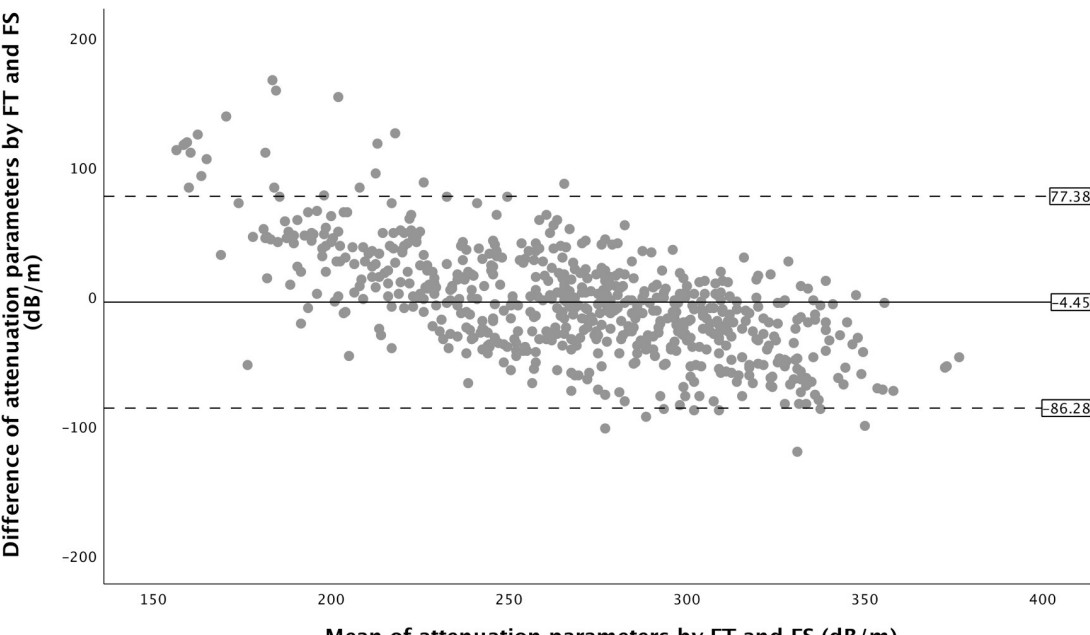

**Fig 2. Bland-Altman plot of attenuation parameters obtained using FibroTouch (FT) and Fibroscan (FS).**

obtained using FT and FS, with the lower 95% limit of agreement being -5.99 kPa and the upper 95% limit of agreement being 7.81 kPa (**Fig 4**). There was substantial agreement between FT and FS with Cohen's kappa value of 0.67 if LSM of 15 kPa was taken as cut-off value. However, if LSM of 10 kPa and 20 kPa were used as cut-offs, there were only moderate agreement with Cohen's kappa value of 0.52 and 0.42, respectively. When the 10 kPa cut-off obtained using FS was used as reference standard for the exclusion of compensated advanced chronic liver disease, LSM obtained using FT had an AUROC of 0.92 for the exclusion of compensated advanced chronic liver disease. The optimal cut-off was 11.7 kPa with sensitivity, specificity, positive predictive value, negative predictive value, and accuracy of 75%, 91%, 69%, 93%, and 88%, respectively. When the 15 kPa cut-off obtained using FS was used as reference

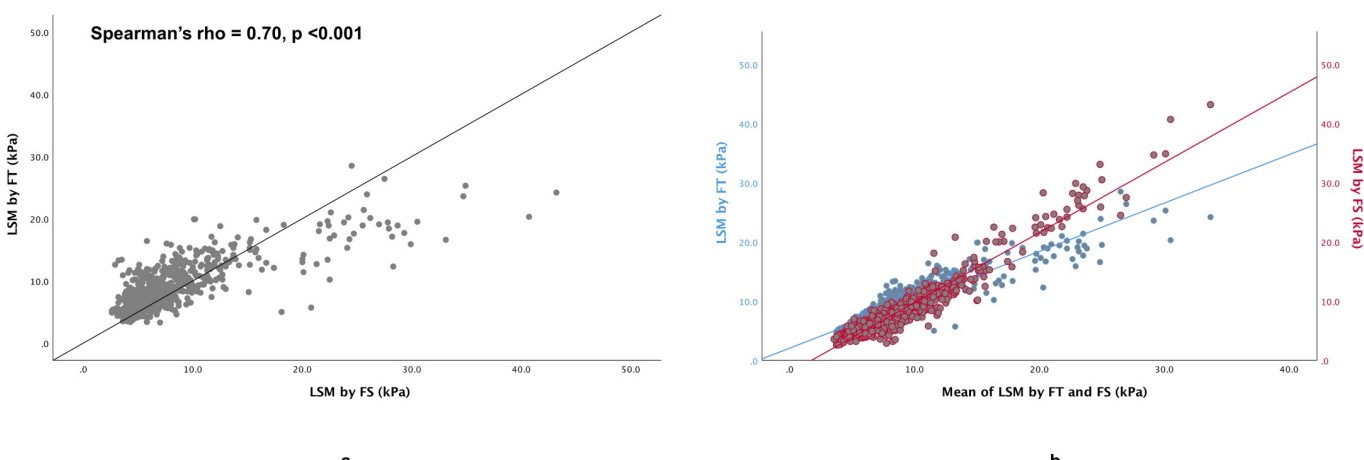

**Fig 3.** Scatter plots for (**a**) liver stiffness measurements (LSM) obtained using FibroTouch (FT) vs Fibroscan (FS), and (**b**) LSM obtained using FT and FS vs mean of LSM obtained using FT and FS.

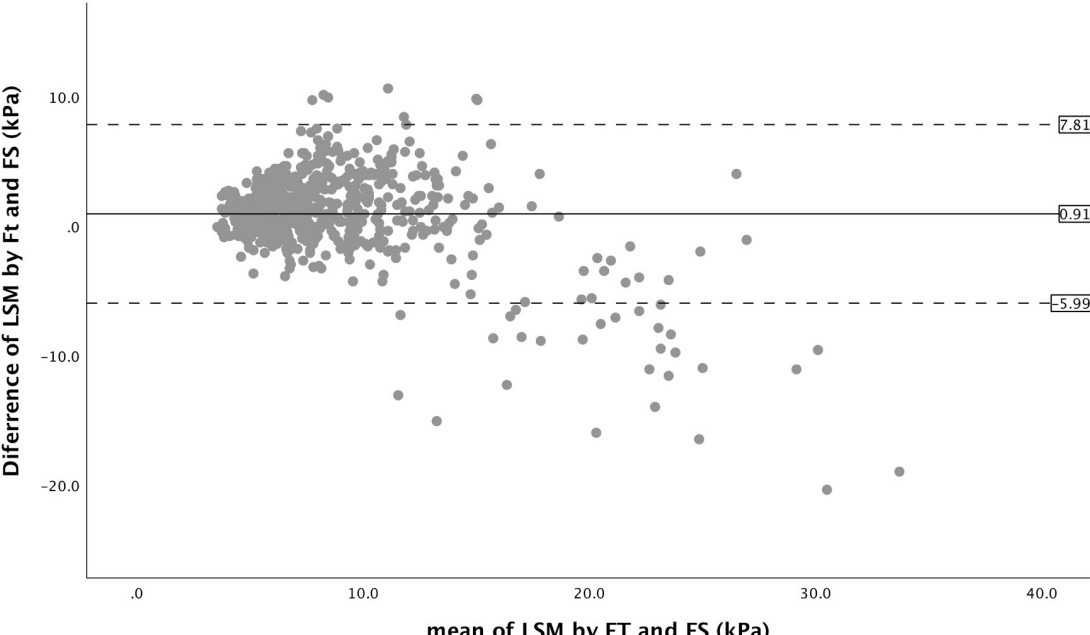

**Fig 4. Bland-Altman plot of liver stiffness measurements (LSM) obtained using FibroTouch (FT) and Fibroscan (FS).**

standard for the diagnosis of compensated advanced chronic liver disease, LSM obtained using FT had an AUROC of 0.93 for the diagnosis of compensated advanced chronic liver disease. The optimal cut-off was 11.8 kPa with sensitivity, specificity, positive predictive value, negative predictive value, and accuracy of 91%, 84%, 35%, 99%, and 85%, respectively. When the 20 kPa cut-off obtained using FS was used as reference standard for the diagnosis of clinically significant portal hypertension, LSM obtained using FT had an AUROC of 0.95 for the diagnosis of clinically significant portal hypertension. The optimal cut-off was 12.7 kPa with sensitivity, specificity, positive predictive value, negative predictive value, and accuracy of 90%, 88%, 33%, 99%, and 88%, respectively.

On subgroup analysis of patients with chronic hepatitis B, if LSM of 8 kPa and 11 kPa were used as cut-offs, there were only moderate agreement with Cohen's kappa value of 0.57 and 0.51, respectively. When the 8 kPa cut-off obtained using FS was used as reference standard for the diagnosis of significant fibrosis, LSM obtained using FT had an AUROC of 0.88 for the diagnosis of significant fibrosis. The optimal cut-off was 8.2 kPa with sensitivity, specificity, positive predictive value, negative predictive value, and accuracy of 93%, 71%, 64%, 95%, and 79%, respectively. When the 11 kPa cut-off obtained using FS was used as reference standard for the diagnosis of cirrhosis, LSM obtained using FT had an AUROC of 0.94 for the diagnosis of cirrhosis. The optimal cut-off was 11.7 kPa with sensitivity, specificity, positive predictive value, negative predictive value, and accuracy of 87%, 89%, 58%, 97%, and 88%, respectively.

On subgroup analysis of patients with NAFLD, if LSM of 7 kPa and 10.3 kPa were used as cut-offs, there were only moderate agreement with Cohen's kappa value of 0.51 and 0.55, respectively. When the 7 kPa cut-off obtained using FS was used as reference standard for the diagnosis of significant fibrosis, LSM obtained using FT had an AUROC of 0.89 for the diagnosis of significant fibrosis. The optimal cut-off was 8.0 kPa with sensitivity, specificity, positive predictive value, negative predictive value, and accuracy of 85%, 81%, 82%, 84%, and 83%, respectively. When the 10.3 kPa cut-off obtained using FS was used as reference standard for the diagnosis of cirrhosis, LSM obtained using FT had an AUROC of 0.91 for the diagnosis of

cirrhosis. The optimal cut-off was 9.8 kPa with sensitivity, specificity, positive predictive value, negative predictive value, and accuracy of 90%, 80%, 56%, 97%, and 82%, respectively.

There was a strong correlation between operator 1 and 2 for LSM measured using FT (Spearman's rho = 0.74, p <0.001) and FS (Spearman's rho = 0.85, p <0.001) (**S2A and S2B Fig**). The intra-observer reliability of LSM using FT by operator 1 and 2 were similar and excellent with ICC 0.97 (95% CI, 0.96–0.98). The intra-observer reliability of LSM using FS by operator 1 and 2 were also excellent with ICC 0.99 (95% CI, 0.98–0.99) and ICC 0.96 (95% CI, 0.94–0.97), respectively. The inter-observer reliability of LSM using FT was good with ICC 0.89 (95% CI, 0.86–0.91), whereas the inter-observer reliability of LSM using FS was excellent with ICC 0.95 (95% CI, 0.94–0.96).

### Time taken for an examination and number of invalid measurements

The time taken for an examination was significantly shorter for FT compared with FS (**Table 3**). The difference remained significant when the analysis was performed separately in obese and non-obese patients. Besides that, FT produced significantly fewer invalid measurements compared with FS. Again, the difference remained significant when the analysis was performed separately in obese and non-obese patients.

## Discussion

In this prospective single centre study on 163 patients with chronic liver disease of various aetiologies, we compared attenuation parameter and LSM obtained using FT and FS. Although there have been several studies comparing FT and FS, all the studies were performed in China, where the FT technology was developed, and five of the six studies were reported in the Chinese literature [17–21]. Only one study has been reported in the English literature (**S1 Table**) [18]. Our study is the first study outside China and the only other study in the English literature comparing FS and FT, thus providing important validation of the findings from earlier studies. Moreover, our study is the only other study comparing attenuation parameter obtained using FT and FS. Our study is also the first study looking at the intra- and inter-observer reliability of attenuation parameter and LSM obtained using FT by healthcare personnel from different background.

**Table 3. Comparison of time taken for an examination and number of invalid measurements between Fibroscan and FibroTouch.**

| | FibroTouch | Fibroscan | p |
|---|---|---|---|
| **Overall** | | | |
| Time taken for an examination, sec | 33 (27–41) | 47 (38–92) | <0.001 |
| Number of invalid measurements | 0 | 2 (1–4) | <0.001 |
| **Non obese** | | | |
| Time taken for an examination, sec | 33 (27–42) | 46 (37–86) | <0.001 |
| Number of invalid measurements | 0 | 1.5 (1–3) | <0.001 |
| **Obese** | | | |
| Time taken for an examination, sec | 32 (26–40) | 50 (38–97) | <0.001 |
| Number of invalid measurements | 0 | 2 (1–5) | <0.001 |

Obesity was defined as BMI $\geq$25 kg per m$^2$.

Three patients who did not have any successful Fibroscan and/or FibroTouch examinations were excluded from the analysis.

**LSM**, liver stiffness measurement.

Our study showed strong correlation between LSM obtained using FT and FS. While this is consistent with previous studies, we noted that the correlation tended to be higher in studies including purely or predominantly patients with chronic hepatitis B [17,18,21,22]. Another study included only patients with primary biliary cholangitis [19]. In the study that included patients with chronic liver disease of various aetiologies [20], the correlation tended to be lower. We also found strong correlation between attenuation parameters obtained using FT and FS. In the only other study comparing attenuation parameter obtained using FT and FS, the correlation was moderate [20]. We believe the differences in correlation between LSM and attenuation parameter obtained using FT and FS in the different studies are likely the result of differences in the performance of the devices in populations with different characteristics. For example, we observed that the diagnostic performance of attenuation parameter is lower in studies with a greater proportion of patients who are obese, have NAFLD or have higher grades of histological hepatic steatosis [23,24]. A subsequent individual patient data meta-analysis confirmed that attenuation parameter can be affected by body mass index and the presence of diabetes mellitus and NAFLD [11].

Interestingly, we observed that FT tended to produce higher LSM compared with FS for lower LSM values and lower LSM compared with FS for higher LSM values. Although there was substantial agreement between FT and FS when LSM of 15 kPa was taken as cut off, there was only moderate agreement between FT and FS if LSM of 10 kPa and 20 kPa were taken as cut off. On the other hand, FT tended to produce higher attenuation parameter compared with FS for lower attenuation parameter values, and lower attenuation parameter compared with FS for higher attenuation parameter values. There was only moderate agreement between FT and FS for diagnosis of the different hepatic steatosis grades. These findings suggest that FT and FS results may not be interchangeable, although they are strongly correlated and may be similarly used for evaluation of patients with chronic liver disease. Moreover, FT appeared to produce more consistent results with significantly lower IQR for attenuation parameter and IQR/median for LSM compared with FS.

Our study revealed high intra- and inter-observer reliability for attenuation parameter and LSM obtained using FT when performed by healthcare personnel of different background, suggesting that FT can be used by healthcare personnel of different background after sufficient training. We have previously published a study comparing FS by healthcare personnel of different background, which showed a similarly high level of intra- and inter-observer reliability [12]. In the current study, we also found that FS produced more invalid measurements and required slightly longer examination time compared with FT. Furthermore, FS has probes of different sizes and the choice of probe is by automatic probe recommendation tool embedded in the FS device. The XL probe was introduced to overcome failed and unreliable examinations using the conventional M probe, especially in obese patients [25]. On the other hand, FT uses one universal probe for examination. The FT probe is equipped with a build-in liver capsule detection module that utilizes the characteristics of internal and external sounds signals of the liver capsule to automatically adjust the depth of examination. In addition to the lower number of invalid measurements, the use of one universal probe may have also contributed to the shorter examination time for FT compared with FS.

Despite our best effort, this study has several limitations. Firstly, liver biopsy was not used as reference standard for the degree of hepatic steatosis and fibrosis in this study. However, there have been numerous studies on the use of FS for estimation of hepatic steatosis and fibrosis using liver biopsy as reference standard. Moreover, FS is established and recognized as a non-invasive tool for these purposes. Therefore, the use of FS as the reference standard for comparison in this study is acceptable. Nevertheless, further studies on FT using liver biopsy as reference standard may be needed in view of our findings of only moderate agreement in

FT and FS diagnosis when applying previously established cut-offs for FS. Secondly, when assessing the inter-observer and intra-observer reliability, the repeat examinations of FT and FS by both operators were performed on the same day. Ideally, these should have been performed at least several days apart to minimize the operator bias, but this was not feasible for logistic reasons. The examinations were performed at different times of the day to minimize operator bias. Thirdly, the results obtained from this study may not be applicable to all aetiologies of chronic liver disease as the majority of the patients had either NAFLD or chronic hepatitis B. Finally, the results of this study may not be applicable to the paediatric population, especially when children have been shown to have age-dependent increases in liver stiffness measurement [26,27]. Further studies are needed to investigate the use of FibroTouch in the paediatric population.

In conclusion, although attenuation parameter and LSM obtained using FT and FS are strongly correlated, there are some differences in values at both extremes of attenuation parameter and LSM values with only moderate agreement in FT and FS diagnosis when applying previously established cut-offs for FS, with the exception of the 15 kPa LSM cut-off, which showed substantial agreement. The intra- and inter-observer reliability of FT and FS are good to excellent suggesting that examination using either devices can be performed by healthcare personnel of different background when they are sufficiently trained. FT does have the advantage of one universal probe with significantly lower invalid measurements and shorter examination time.

## Supporting information

**S1 Fig.** Scatter plots of attenuation parameters obtained by operator 1 vs operator 2 using (a) Fibrotouch, and (b) Fibroscan.
(TIF)

**S2 Fig.** Scatter plots of liver stiffness measurements obtained by operator 1 vs operator 2 using (a) Fibrotouch, and (b) Fibroscan.
(TIF)

**S1 Table. Summary of studies that evaluated the performance of Fibroscan (FS) and Fibro-Touch (FT).**
(DOCX)

**S1 File. Complete dataset to replicate all of the figures, tables, statistics, and other values in this paper.**
(SAV)

## Acknowledgments

We would like to thank Wellmedic Healthcare Pvt. Ltd. by providing the FibroTouch device used in this study. The company was not involved in the study design; in the collection, analysis and interpretation of data; in the write up of the report; and in the decision to submit the paper for publication. An abstract for this work was submitted to The Liver Meeting Digital Experience 2020, where it was presented as a poster, and the abstract was published in the corresponding supplementary issue of Hepatology.

## Author Contributions

**Conceptualization:** Wah Kheong Chan.

**Data curation:** Ying Zhuang Ng, Lee Lee Lai, Sui Weng Wong, Siti Yatimah Mohamad, Kee Huat Chuah.

**Formal analysis:** Ying Zhuang Ng.

**Validation:** Wah Kheong Chan.

**Writing – original draft:** Ying Zhuang Ng.

**Writing – review & editing:** Wah Kheong Chan.

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
