## [Decision Letter · Decision Letter 0]

22 Dec 2020

PONE-D-20-33839

Attenuation parameter and liver stiffness measurement using FibroTouch vs Fibroscan in patients with chronic liver disease

PLOS ONE

Dear Dr. Chan,

Thank you for submitting your manuscript to PLOS ONE. After careful consideration, we feel that it has merit but does not fully meet PLOS ONE’s publication criteria as it currently stands. Therefore, we invite you to submit a revised version of the manuscript that addresses the points raised during the review process.

We look forward to receiving your revised manuscript.

Kind regards,

Daisuke Tokuhara

Academic Editor

PLOS ONE

Additional Editor Comments:

Thank you for submitting your valuable study. Transient elastography is the non-invasive device for the evaluation of chronic liver disease thus its application will be popular worldwidely. Therefore, for the physicians, it is important to know the usefulness, simillarities and differences of FibroTouch and FibroScan. Two reviewers reviewed strictly and provided the constructive comments. I believe those comments will contribute to improve the manuscript. Please reply to their comments thoroughly.

Journal Requirements:

3. Thank you for submitting the above manuscript to PLOS ONE. During our internal evaluation of the manuscript, we found text overlap between your submission, specifically your abstract, and the following previously published work.

- https://aasldpubs.onlinelibrary.wiley.com/doi/10.1002/hep.31579

Before we proceed, would you please kindly clarify if the published abstract was previously peer-reviewed? If so, please explain in your cover letter why this work does not constitute a dual publication.

Please note that should your paper be accepted, all content including images will be published under the Creative Commons Attribution (CC BY) 4.0 license, which means that they will be freely available online, and any third party is permitted to access, download, copy, distribute, and use these materials in any way, even commercially, with proper attribution. In order to publish any previously copyrighted material, PLOS ONE requires permission from the original copyright holder of the content to publish it under the CC BY 4.0 license.

Please clarify whether the authors have received written permission from Wiley to publish this content specifically under the CC BY 4.0 license and upload the granted permission to the manuscript as a supporting information file.

Reviewers' comments:

Reviewer's Responses to Questions

**Comments to the Author**

1. Is the manuscript technically sound, and do the data support the conclusions?

Reviewer #1: Yes

Reviewer #2: Yes

2. Has the statistical analysis been performed appropriately and rigorously? 

Reviewer #1: Yes

Reviewer #2: Yes

3. Have the authors made all data underlying the findings in their manuscript fully available?

Reviewer #1: No

Reviewer #2: Yes

4. Is the manuscript presented in an intelligible fashion and written in standard English?

Reviewer #1: Yes

Reviewer #2: Yes

5. Review Comments to the Author

Reviewer #1: The authors have performed a meticulous study to compare results of FS with FT across a spectrum of etiologies and disease severities. However some issues remain:

In abstract it is mentioned that there were 0 invalid measurements with FT vs 2 with FS, but in results it is mentioned that there were 2 failed with FS alone, and 1 with FS and FT both, meaning 3 invalid with FS and 1 with FT. Please clarify.

While correlation between FS and FT is good, the absolute values in a given patient are quite significantly different both for attenuation parameter and LSM specially for cases in the lowest and highest ranges. Hence cut-offs for FS and FT cannot be same. FS has well validated cut-off, but FT cut-offs need to be validated with liver biopsies.

The authors have used LSM cut-offs of <10, >15 and >20 in this study for absence of cirrhosis, presence of cirrhosis and presence of significant portal HTN. But this difference is usually apparent after USG and endoscopy. In real life we often use further refined cut-offs between 6 and 15KPa to separate out various stages of fibrosis from F0 to F4 since it is necessary to determine whether a given patient has significant or advanced fibrosis or not without resorting to liver biopsy. Comparison of FS and FT using these cut-offs would be clinically more relevant. As per my experience FT tends to have higher LSM values in this range.

Taking validated FS cut-offs as the gold standard (since it is better validated than FT), the sensitivity, specificity and accuracy of FT for predicting various stages of fibrosis as well as for predicting portal hypertension should be calculated to make this study more clinically meaningful.

Reviewer #2: The current study strictly compare the parameters between FibroScan and FibroTouch. I have no comments on the data and the data-based conclusions. But there is a limitation should be described adn discussed. FibroScan provided S, M and XL probe, thus S probe can be applied to children. In addition, children has age-dependent different reference value in LSM that was disclosed by the accumulated studies (Liver International. 2020;40:2602-2611; PLoS One. 2016;11:e0166683). On the other hand, FibroTouch has only one type of probe. I think FibroTouch's probe will not be applied to the narrow intercostal space of the children. Or, a future study is required for the FibroTouch to evaluate the applicapability in children. Authors are required to include the discussion or limitation of the study regarding the above point (applicapability in children) by citing the recent above references.

6. PLOS authors have the option to publish the peer review history of their article (what does this mean?). If published, this will include your full peer review and any attached files.

Reviewer #1: **Yes: **Swastik Agrawal

Reviewer #2: No

---

## [Author Response · Author response to Decision Letter 0]

18 Mar 2021

18th March 2021 

Professor Daisuke Tokuhara

Academic Editor

PLOS ONE

Ms. Ref. No.: PONE-D-20-33839

Title: Attenuation parameter and liver stiffness measurement using FibroTouch vs Fibroscan in patients with chronic liver disease

We would like to thank the editor and reviewers for the invaluable comments that have helped to improve the quality of our manuscript. Following are the point-to-point responses to the comments: 

Reviewer #1: The authors have performed a meticulous study to compare results of FS with FT across a spectrum of etiologies and disease severities. However some issues remain:

In abstract it is mentioned that there were 0 invalid measurements with FT vs 2 with FS, but in results it is mentioned that there were 2 failed with FS alone, and 1 with FS and FT both, meaning 3 invalid with FS and 1 with FT. Please clarify.

Reply: We regret that these details were not clear in the originally submitted manuscript. The FS examination was unsuccessful (i.e., <10 valid measurements were obtained after 30 attempts) all four times (examination was performed two times by each of the two operators) in three patients. The FT examination was unsuccessful (i.e., <10 valid measurements were obtained after 30 attempts) all four times (examination was performed two times by each of the two operators) in one patient (who also had unsuccessful FS all four times). Comparison cannot be made without any successful FS and/or FT examination. Therefore, these three patients were excluded from the analysis. On the other hand, the 0 invalid measurements for FT and median 2 (IQR 1 - 4) invalid measurements for FS (presented in Table 3) are the average numbers of invalid measurements for the examinations performed on the 163 patients included in the analysis. The three patients who did not have any successful Fibroscan and/or FibroTouch examinations were excluded from the analysis. We have included a note on this below Table 3 for clarity. In addition, we have revised the abstract for clarity by stating the number of patients and examinations that were included in the analysis instead of the number of patients who were enrolled in the study and the number of attempted examinations. We have also explicitly state that 163 patients with 1304 examinations were included in the analysis in the results section of the revised manuscript. 

While correlation between FS and FT is good, the absolute values in a given patient are quite significantly different both for attenuation parameter and LSM specially for cases in the lowest and highest ranges. Hence cut-offs for FS and FT cannot be same. FS has well validated cut-off, but FT cut-offs need to be validated with liver biopsies.

Reply: Thank you for this comment. We completely agree and have already mentioned similar points in the discussion section of the originally submitted manuscript, albeit in separate paragraphs.

“These findings suggest that FT and FS results may not be interchangeable, although they are strongly correlated and may be similarly used for evaluation of patients with chronic liver disease.”

“Nevertheless, further studies on FT using liver biopsy as reference standard may be needed in view of our findings of only moderate agreement in FT and FS diagnosis when applying previously established cut-offs for FS.”

The authors have used LSM cut-offs of <10, >15 and >20 in this study for absence of cirrhosis, presence of cirrhosis and presence of significant portal HTN. But this difference is usually apparent after USG and endoscopy. In real life we often use further refined cut-offs between 6 and 15KPa to separate out various stages of fibrosis from F0 to F4 since it is necessary to determine whether a given patient has significant or advanced fibrosis or not without resorting to liver biopsy. Comparison of FS and FT using these cut-offs would be clinically more relevant. As per my experience FT tends to have higher LSM values in this range.

Reply: Thank you for this comment. Previous studies on different populations have found different liver stiffness measurement cut-offs for the diagnosis of different fibrosis stages. Importantly, there is substantial overlap in liver stiffness measurement between adjacent fibrosis stages, and the selection of cut-offs is based on trade-off between sensitivity and specificity. To simplify matters, the Baveno VI Consensus Workshop has recommended liver stiffness measurement <10 kPa to rule-out compensated advanced chronic liver disease, 10-15 kPa to be suggestive of compensated advanced chronic liver disease, >15 kPa to be highly suggestive of compensated advanced chronic liver disease, and ≥20-25 kPa to be having clinically significant portal hypertension (PMID: 26047908). We have also validated the 10 kPa and 15 kPa cut-offs in our study on NAFLD patients (PMID: 30658997). Therefore, we have used these cut-offs in our current study. However, in response to this comment, we have included subgroup analysis for patients with chronic hepatitis B infection and for patients with NAFLD. The number of patients with chronic liver disease of other aetiologies is too small for subgroup analysis. We have used the 8 kPa and 11 kPa cut-offs for significant fibrosis and cirrhosis, respectively, for patients with chronic hepatitis B infection (PMID: 26563120), and the 7.0 kPa and 10.3 kPa cut-offs for significant fibrosis and cirrhosis, respectively, for patients with NAFLD (PMID: 20101745). We have included the results in the revised manuscript, updated the methods section and cited the references accordingly. 

Taking validated FS cut-offs as the gold standard (since it is better validated than FT), the sensitivity, specificity and accuracy of FT for predicting various stages of fibrosis as well as for predicting portal hypertension should be calculated to make this study more clinically meaningful.

Reply: Thank you for this comment. We have included the AUROC, optimal cut-off, sensitivity, specificity, positive predictive value, negative predictive value, and accuracy of FT using validated FS cut-offs as the gold standard in the revised manuscript as suggested and updated the methods section accordingly. 

Reviewer #2: The current study strictly compare the parameters between FibroScan and FibroTouch. I have no comments on the data and the data-based conclusions. But there is a limitation should be described adn discussed. FibroScan provided S, M and XL probe, thus S probe can be applied to children. In addition, children has age-dependent different reference value in LSM that was disclosed by the accumulated studies (Liver International. 2020;40:2602-2611; PLoS One. 2016;11:e0166683). On the other hand, FibroTouch has only one type of probe. I think FibroTouch's probe will not be applied to the narrow intercostal space of the children. Or, a future study is required for the FibroTouch to evaluate the applicapability in children. Authors are required to include the discussion or limitation of the study regarding the above point (applicapability in children) by citing the recent above references.

Reply: Thank you for this comment. We have included these points in the discussion section of the revised manuscript and cited the references accordingly.

“Finally, the results of this study may not be applicable to the paediatric population, especially when children have been shown to have age-dependent increases in liver stiffness measurement. Further studies are needed to investigate the use of FibroTouch in the paediatric population.” 

In addition, we have written non-alcoholic fatty liver disease in full when it appeared in the manuscript for the first time. We have also made some edits to the structure of several sentences in the abstract so that it is less similar to the abstract of this work that was submitted for poster presentation at The Liver Meeting Digital Experience 2020 and subsequently published in the supplementary issue of Hepatology. We have obtained approval from Wiley to publish the amended abstract along with this full text of our work, which has not been published elsewhere before. 

Once again, we would like to thank the editor and reviewers for the comments put forth to help us improve on our manuscript. We have tried our very best to answer to the comments and to make all the necessary changes to the manuscript and hope that it will be acceptable to the editors and reviewers.

Thank you very much and looking forward to hearing from you!

Best regards, 

Dr Chan Wah Kheong

Professor and Senior Consultant Gastroenterologist and Hepatologist

Department of Medicine, Faculty of Medicine, University of Malaya, 50603 Kuala Lumpur, Malaysia

Telephone no.: +60379492965

Fax no.: +60379604190

E-mail: wahkheong2003@hotmail.com

---

## [Decision Letter · Decision Letter 1]

5 Apr 2021

Attenuation parameter and liver stiffness measurement using FibroTouch vs Fibroscan in patients with chronic liver disease

PONE-D-20-33839R1

Dear Dr. Chan,

We’re pleased to inform you that your manuscript has been judged scientifically suitable for publication and will be formally accepted for publication once it meets all outstanding technical requirements.

Kind regards,

Daisuke Tokuhara

Academic Editor

PLOS ONE

Additional Editor Comments (optional):

Authors well addressed to the comments from reviewers. I appreciate the authors's continuous efforts for revision.

Reviewers' comments:

Reviewer's Responses to Questions

**Comments to the Author**

1. If the authors have adequately addressed your comments raised in a previous round of review and you feel that this manuscript is now acceptable for publication, you may indicate that here to bypass the “Comments to the Author” section, enter your conflict of interest statement in the “Confidential to Editor” section, and submit your "Accept" recommendation.

Reviewer #1: All comments have been addressed

Reviewer #2: All comments have been addressed

2. Is the manuscript technically sound, and do the data support the conclusions?

Reviewer #1: Yes

Reviewer #2: Yes

3. Has the statistical analysis been performed appropriately and rigorously? 

Reviewer #1: Yes

Reviewer #2: Yes

4. Have the authors made all data underlying the findings in their manuscript fully available?

Reviewer #1: Yes

Reviewer #2: Yes

5. Is the manuscript presented in an intelligible fashion and written in standard English?

Reviewer #1: Yes

Reviewer #2: Yes

6. Review Comments to the Author

Reviewer #1: (No Response)

Reviewer #2: Authors satisfactorilly responded to the suggestions and comments from reviewers. I have no further comments to the revised manuscript.

7. PLOS authors have the option to publish the peer review history of their article (what does this mean?). If published, this will include your full peer review and any attached files.

Reviewer #1: **Yes: **Swastik Agrawal

Reviewer #2: No

---

## [Editor Report · Acceptance letter]

23 Apr 2021

PONE-D-20-33839R1 

Attenuation parameter and liver stiffness measurement using FibroTouch vs Fibroscan in patients with chronic liver disease 

Dear Dr. Chan:

I'm pleased to inform you that your manuscript has been deemed suitable for publication in PLOS ONE. Congratulations! Your manuscript is now with our production department. 

Kind regards, 

on behalf of

Dr. Daisuke Tokuhara 

Academic Editor

PLOS ONE